# Burnout and Quality of Work Life among Municipal Workers: Do Motivating and Economic Factors Play a Mediating Role?

**DOI:** 10.3390/ijerph192013035

**Published:** 2022-10-11

**Authors:** Dina Pereira, João Leitão, Ludovina Ramos

**Affiliations:** 1Centre for Management Studies of Instituto Superior Técnico (CEG-IST), University of Lisbon, 1049-001 Lisboa, Portugal; 2Research Center in Business Sciences (NECE), University of Beira Interior, 6200-209 Covilhã, Portugal; 3Faculty of Human and Social Sciences, University of Beira Interior, 6200-001 Covilhã, Portugal; 4Instituto de Ciências Sociais (ICS), University of Lisbon, 1649-004 Lisboa, Portugal

**Keywords:** burnout, occupational health, public sector, quality of work life, well-being

## Abstract

This study analyzes the relationship between burnout and quality of work life among municipal workers subjected to higher levels of stress and emotional exhaustion, impacting their occupational health in the context of the COVID-19 pandemic. With a sample of 459 municipal workers, the relationship between burnout and quality of work life is tested by considering the isolated mediating effect of the feeling of contributing to productivity and the combined effects of two mediators representing the feeling of contributing to productivity and receiving an appropriate salary. The main findings include a negative association between the three dimensions of burnout: emotional exhaustion, feelings of cynicism, and a sense of being less effective, and the mediators: contribution to productivity and appropriate salary. Also detected was an important mediating role associated with the effects of not feeling contributive at work, as well as not being well paid, on the relation between the burnout syndrome dimension of low effectiveness and quality of work life. For future action by public authorities and public managers, the need is highlighted to create innovative human resource management frameworks and flexible work organization, with remuneration plans based on productivity goals and aimed at an improved balance between personal life and work.

## 1. Background

The concepts of quality of work life (QWL) and job satisfaction are interrelated, when attempting to define a good work environment or a good and healthy life. Sirgy et al. [1] state that QWL relates to job satisfaction in the sense that job satisfaction is one of many outcomes of QWL. Job satisfaction is crucial because it influences job performance, customer satisfaction, employment retention, employee absenteeism, and organizational commitment [2]. Moreover, QWL does not only affect job satisfaction but also satisfaction in other life domains, such as family life, leisure life, social life, and financial life. In fact, women working from home during the COVID-19 lockdown experienced a set of negative effects from working remotely, namely the invasion of privacy, family time, the occurrence of distractions while working resulting in failures, and the increasing interference of work in private life [3]. Therefore, the focus of QWL goes beyond job satisfaction. It involves the effect of the workplace on satisfaction with the job; satisfaction in non-work life domains; and satisfaction with overall life, personal happiness, and subjective well-being [1].

Work occupies one’s thoughts, determining decisions and thus contributing to one’s social identity [4]. Such subjective and behavioral components of the QWL, namely supervisors’ support, a good work environment, and collaborative support from co-workers, as well as the feeling of being respected professionally and personally, have an important influence on forming an employee’s individual desire to contribute to the organization’s productivity. This is also supported by the statement that QWL is associated with job satisfaction, motivation, productivity, health, job security, safety, and well-being, embracing four main axes: a safe work environment, occupational healthcare, appropriate working time, and an appropriate salary [5,6].

With the increasing workloads of the past decades, plus the COVID-19 pandemic, the number of employees experiencing psychological problems related to occupational stress has increased rapidly, raising costs in terms of absenteeism and loss of productivity and also increasing healthcare consumption and raising public health issues in the long term [7]. Conversely, not only at the employees’ level, but also at the level of SME owners, who faced, during the COVID-19 crisis, high stress, mostly derived from the shortage of personnel, financial constraints, liquidities, repeated closures, and reopenings, as well as great difficulty in adapting to such a changing environment [8].

Added to the above, occupational stress and self-reported sleep quality are also strongly associated with both QWL and work ability, highlighting the urgent need for the screening and handling of these health issues [9]. Occupational stress leads to organizational burnout, whose effects were formerly analyzed as moderators of the relationship between employees’ QWL and their perceptions of their contribution to the organization’s productivity by integrating the QWL factors into the trichotomy of (de)motivators of productivity in the workplace [10]. Our previous findings suggest that QWL hygiene factors (e.g., safe work environment and occupational healthcare) have an important influence on productivity, and burnout de-motivator factors (that is, low effectiveness, cynicism, and emotional exhaustion) significantly moderate the relationship between QWL and the contribution to productivity.

The COVID-19 pandemic—which has been (and will continue to be) a key issue throughout and beyond the 2020s—had an effect on individual–organizational relations and consequently, on organizational performance [11]. This took a toll not only on frontline workers (physicians, nurses, and hospital workers) but also on other public servants, such as municipal workers, not studied so far, who were affected by the changing the location of their work, tasks, the demands at work, and the demands they face outside work, endangering the balance between professional and personal life [12,13].

Furthermore, the financial insecurity and financial stress that already interfered with work [14] increased unexpectedly as COVID-19 aggravated these concerns. Due to the COVID-19 pandemic, employees are generally more aware of financial security. According to Kulikowski and Sedlak (2020), studies on work engagement and job performance have shown that employees ranked financial security as a factor of the highest significance [15]. These concerns could trigger stress, impacting employees’ health and leading to mental illnesses such as post-traumatic stress disorder (PTSD) [13] but also high rates of tension, anger, anxiety, depressed mood, mental fatigue, and sleep disturbances. Such problems, usually referred to overall as distress, are often classified as neurasthenia, adjustment disorders, or burnout [7,16]. Focusing on younger, unmarried female healthcare workers with low monthly incomes who were studied in Jordan during the pandemic, they revealed high anxiety symptoms, which were exacerbated one year after the beginning of COVID-19, particularly for the physicians, with intense schedules, and those who were infected [17]. This situation is intensified in the public healthcare system, during pandemics, as it is characterized by a scarcity of resources and reduced accountability, thus increasing the lack of trustworthiness in the health system [18]. Strategies to improve healthcare systems’ efficiency, physicians’ motivation, management routines, patient flows, and information are a need during extreme crises, such as COVID-19. These strategies were implemented in a nodal-designated COVID-19 center in Qatar and proved to be effective, lowering mortality and increasing efficacy, capabilities, and patient satisfaction [19].

Furthermore, employees’ mental health has a positive relationship with job performance, although this same relationship is mediated by innovative behavior and work engagement, which are also positively associated with job performance [20].

Knowledge on the influencers of QWL has remained relatively limited. In order to forecast and make possible an anticipated action on the part of responsible managers, it is essential to assess hypothetical unexplored effects associated with other mediating factors of the core relationship between burnout and QWL [21].

Set against this background, a relevant research question arises concerning the need to advance the still limited knowledge about other types of motivating and economic mediators affecting municipal workers’ QWL, especially in terms of disorders associated with burnout, in the context of the COVID-19 pandemic. An effective response to the COVID-19 pandemic required effective administration, which in turn depended on the effort and capacity of millions of public sector workers from the front line to central administration. Added to the above, for many public servants, COVID-19 has fundamentally changed not only where and how they work but also the increasing demands of their jobs and day-to-day life [12].

## 2. Burnout during COVID-19 for Public Servants

When the World Health Organization (WHO) declared the outbreak of a new coronavirus disease (COVID-19) and then characterized it as a pandemic, all working people were affected. Not only because of their (or their relatives’) health, but also due to the implications for the workplace. Here, we can identify three main groups of people who could be affected by burnout, namely: (a) people who kept their jobs and continued to work in the same location, (b) people who kept their jobs but started to work from home, and (c) people who lost their jobs.

The term “burnout” was first used by Freudenberger (1974), who described his own experience as “a combination of feelings, exhaustion and fatigue, a lingering cold, headache and gastrointestinal disturbances, sleeplessness and shortness of breath” [22]. The discussion on burnout has grown since then, being continuously updated in terms of symptoms but also in terms of consequences, not only for the employee, but also for the employer.

The multidimensional theory of burnout [23,24,25] defines burnout as being grounded in three core components: (i) emotional exhaustion, as the individual stress dimension of burnout, refers to energy depletion or the draining of emotional resources; (ii) depersonalization, which represents the interpersonal dimension, refers to the development of negative, cynical attitudes towards the recipients of one’s service or care; and (iii) reduced or lack of personal accomplishment, which refers to a decline in one’s feeling of competence and successful achievement in one’s career; this is the self-evaluation dimension of burnout. Furthermore, burnout is characterized by: (a) loss of enthusiasm for work; (b) psychological exhaustion; (c) indolence, and the appearance of negative attitudes and behaviors towards patients and the organization; and (d) the appearance, in some cases, of feelings of guilt [26].

Globally, burnout entails a state of physical, emotional, and mental exhaustion resulting from a long period of involvement in highly emotional demanding work situations [27]. Burnout is a psychological syndrome of exhaustion, cynicism, and inefficacy in the workplace. It is also considered to be an individual stress experience embedded in a context of complex social relationships, involving a person’s perception of both the self and others on the job [28,29], associated with limited resources, low abilities, and low energies and interest in long-term work [30]. This condition involves emotional exhaustion, depersonalization, and a lack of personal accomplishment. Burnout can arise through stress disorders triggered by stress on the job and is also often defined as an emotional-exhaustion experience by employees [15,31]. Theoretical frameworks such as the JD-R model claim that high, unfavorable job demands are consistently related to burnout [32,33,34,35,36]. Moreover, the combination of those high demands with low job resources can lead to some kind of long-lasting burnout and employee disengagement.

As a phenomenon that is context-specific, it is worthwhile to try to deepen the still-scarce knowledge about the impact of pandemic circumstances on occupational health, insofar as it has influenced working conditions, involvement, QWL, and burnout levels, bearing in mind the limited knowledge about the situation for public servants, including municipal workers. According to Meyer et al. (2021, p. 1) [37], “(…) the COVID-19 pandemic poses new challenges for employees’ psychological health that go beyond previous findings in the area of demands and resources (e.g., [35,38]).

The spread of COVID-19, followed by swift responses by companies and governments, created many new challenges that have brought about profound changes and affected the normal health routines and lifestyles of people of all ages, restricting outdoor or physical activity, increasing sedentary time, and consequently, disrupting sleep [39]. Nevertheless, the same authors outline that the isolation of workers at home had mixed effects on adult health behaviors in China, stressing that those workers focused more on their eating quality and patterns, which had a positive influence on their quality of life. The biggest change for most workers was the remote work experience. Prior to COVID-19, most workers had little remote working experience, nor were they or their organizations prepared to adopt this practice. Now, the unprecedented pandemic has required millions of people across the world to become remote workers, inadvertently leading to a de facto global experiment on remote working [40,41]. According to a study by Moretti et al. (2001) of the home-working population, home workers perceived themselves to be less productive compared to their office working period and less satisfied due to isolation [42].

In this respect, COVID-19 highlights employees’ and employers’ vulnerability. As many businesses around the world will be restructured or disappear due to the pandemic, workers will be retrained or laid off and the economic, social–psychological, and health costs of these actions are likely to be immense. Indeed, the impacts of the pandemic affect some groups of workers more than others, for example, based on their age, race and ethnicity, gender, or personality [41]. These impacts were affirmed by Wang et al. (2021), who stated that, as schools in China shut down during the COVID-19 outbreak, working parents faced a challenge in balancing work and family roles and time, creating higher levels of exhaustion, depression, and burnout [40].

Blake at al. (2020) conducted a survey of frontline/healthcare workers to understand the psychological impacts on employees and how these translated into negative consequences for organizations [43]. They found that the extreme pressure experienced by workers during the COVID-19 pandemic might increase their risk of burnout, which has adverse outcomes, not only for their individual well-being, but also for patient care and for the healthcare system. In addition, fear of exposure to COVID-19 or even due to the scarcity of personal protective equipment (PPE), allocation of resources, accountability, and the efficient management of patient flows, as well as a lack of or a reduction in training on practical skills to deal with emergency situations and critical care, can put even more pressure on frontline/health professionals [18,44]. There is also the fear of taking the disease home or even being responsible for bringing it to the workplace, infecting other patients. All this combined with the normal challenges of supporting a family, changes in workload and schedules, and facing new or unknown clinical situations may substantially increase levels of anxiety, emotional strain, and physical exhaustion.

At the coalface of the pandemic, healthcare workers and public service providers have jobs and occupations that have proven to be associated with increased mental health problems during pandemic crises and high personal and work-related burnout [13].

A survey applied to Portuguese healthcare workers during the COVID-19 pandemic, wherein frontline working positions were associated with higher levels of stress and depression, showed a significant association with increased burnout levels. In this survey, higher levels of satisfaction with life and resilience were highly associated with lower levels of burnout [45]. This is also supported in the research developed by Hofmam and Hubie (2020), wherein surveys conducted among frontline workers from two health units in Cascavel-PR (Brazil) identified that all participants obtained higher than expected burnout scores, emotional exhaustion, and a feeling of low professional achievement [46].

Pandemics bring new ways of performing jobs in several sectors of activity, including among public servants not directly working in health-related areas, as happens with municipal workers. The institutionalization of remote working and the rapid, sometimes reckless, digitalization by companies and public organizations has increased the frequency of workers’ burnout with consequences for families’ personal life and budget [47].

During the COVID-19 pandemic, municipal workers working in sectors closer to citizens were forced to work remotely, which has caused unique supervisory demands for human resources managers [12], as well as exacerbated burnout caused by self-isolation policies, which can increase social isolation and relationship difficulties [48].

Other public sector activities and their workers, for instance, those connected with education, were also put under pressure by the effects of pandemics. Marelli et al. (2021) showed a high percentage of both students and university administration staff workers denoting symptoms of depression or anxiety [49], and Evanoff et al. (2020) pointed out the important prevalence of stress, anxiety, depression, work exhaustion, burnout, and worsened well-being among university employees [50]. Another study on insomnia among employees in occupations critical to the functioning of society (e.g., health, education, welfare, and emergency services) during the COVID-19 pandemic also found that employees reported higher levels of insomnia symptoms compared to normative data collected before the pandemic [51]. These findings only highlight the associations of health and well-being with additional personal and work factors beyond the COVID-19 pandemic, without addressing the role played by motivating and economic mediators of the relationship between burnout and QWL.

A study of social workers in the United States during the COVID-19 pandemic, showed a high level of PTSD and burnout symptoms among the participants enrolled in frontline jobs dealing directly with the risk of contagion [52]. Bapuji et al. (2020) stated that organizational and societal inequalities feed into each other, giving rise to concerns that growing inequality after COVID-19 will also contribute to a downward spiral of negative trends in the workplace in the form of decreased work centrality and increased burnout, absenteeism, deviant behaviors, bullying, and higher job rotation [53].

## 3. Methods

### 3.1. Research Question and Model Design

The aim of this study is to address the following core research question: Does the feeling of contributing to productivity and receiving an appropriate salary mediate the relationship between the burnout condition and the QWL of municipal workers in the context of the COVID-19 pandemic?

To address this research question, the model specification presented in Figure 1 below intends to evaluate the relationship between three burnout dimensions and QWL, mediated by motivating and economic factors, including both the feeling of contributing to productivity and receiving an appropriate salary, regarding municipal workers.

### 3.2. Study Design and Participants

This cross-sectional quantitative study examined municipal workers in Portugal. The survey was administered by emailing the questionnaire to Portuguese municipalities, from December 2020 to April 2021, with 459 responses being received from municipal workers developing professional activities during the COVID-19 pandemic, with a diversified set of socio-economic characteristics and professional profiles, in municipal institutions with different sizes, as presented below in Table 1.

### 3.3. Survey Procedures

The research methodology was developed using different surveys, which were designed taking a set of eleven international benchmarks into consideration, namely: (i) health and well-being at work: a survey of employees, 2014, UK, Department for Work and Pensions [54]; (ii) ACT Online Employee Health and Wellbeing Survey 2016, Australian Capital Territory Government [55]; (iii) British Heart Foundation 2012, employee survey [56]; (iv) British Heart Foundation 2017, staff health and wellbeing template survey [57]; (v) Rand Europe (2015), Health, wellbeing and productivity in the workplace—Britain’s Healthiest Organization summary report [58]; (vi) South Australia Health, Government of South Australia staff needs assessment, staff health and wellbeing survey; (vii) Southern Cross Health Society and Business NZ, Wellness in the Workplace Survey 2017 [59]; (viii) State Government Victoria, Workplace Health & Wellbeing needs survey; (ix) East Midlands Public Health Observatory, Workplace Health Needs Assessment for Employers, February 2012 [60]; (x) Tool for Observing Worksite Environments (TOWE). U.S. Department of Health & Human Services [1]; and (xi) Measure of QWL, as originally proposed in [61].

With the motivation of accomplishing the objectives, this study was analytical and correlational, because it sought to explore the variables and the relationships between them, and it was cross-sectional because the sample was collected in a single period. The purpose of the study was descriptive because it aimed to analyze the relationship between burnout and quality of work life among municipal workers subjected to higher levels of stress and emotional exhaustion, interacting with their occupational health in the context of the COVID 19 pandemic. Through a quantitative, objectivist, and, therefore, deductive approach, this research was supported by models built on results and previous research, with quantitative indicators collected through a survey.

The survey includes four sections: (i) health, (ii) well-being, (iii) QWL, and (iv) sample characterization (gender, age, marital status, role in the organization, type of employee contract, academic qualifications, size of the organization). In the first three sections, Likert scales were used (ranging from 1 to 7), in order to evaluate the level of agreement with a set of affirmations, which were used to assess the level of agreement with a set of sentences in each sub-section. These scales had been transformed into binary considering the variables under analysis. In the fourth section, levels of answers were used, considering values ranging from 1 to 4 equaling 0 and values 5, 6, and 7 equal to 1.

The study period included a declaration of national calamity twice and two subsequent stages of easing lockdown measures following the states of national emergency (between 19 March and 4 May 2020 and between 15 January and 15 March 2021). A questionnaire built in the Google platform was provided to participants via a link shared through direct e-mail to municipalities’ contacts.

### 3.4. Measures and Covariates

Socio-demographic and other mental health related data were collected through a self-administered survey. The variables used in to measure burnout were: emotional exhaustion, cynicism, and low effectiveness [8]. All items were scored on a 7-point Likert scale. The variables used to measure QWL were: supervisor support, co-worker support, good work environment, professional respect, work–life balance, and skills development [4]. QWL was then generated as a new variable, computed by using the six prior variables, attributing the value 1 if the levels of agreement were positive and 0 otherwise. Regarding the mediators, the variables of contribution to productivity and appropriate salary were used. The former was scored on a 7-point Likert scale, ranging from 1 if the worker felt they contribute to the organization’s productivity, 0 otherwise, while the latter ranged from 1 to 7, 1 being if the workers did not agree and 7 if they totally agreed.

### 3.5. Data Analysis

The data collected from Google^®^ Forms were exported to an Excel spreadsheet, and all statistical analyses were carried out using STATA Statistics (version 14.1). Variable characterization was performed by means of absolute and relative frequencies, means and standard deviations (SDs).

Two mediation analyses were performed using two models. Model 1 used the intermediate variable (contribution to productivity), and Model 2 used two intermediate variables (contribution to productivity and appropriate salary). These mediators intend to explain how or why a set of independent variables influences an outcome (here the QWL). To do so, structural equation modeling (SEM) was used, as it is a powerful multivariate technique, which makes use of a conceptual model, path diagram, and system of linked regression-style equations in order to capture complex and dynamic relationships within an agglomerate of both observed and unobserved variables. This technique allows a reciprocal role played by a variable and enables the inference of causal relationships. According to Gunzler et al. (2013), the option for the SEM framework in a mediation analysis is advantageous when the model comprises latent variables such as quality of life or stress, as it will make interpretation and estimation easier [61]. As SEM was created partly to test complex mediation models in a single analysis, this technique simplifies the testing of mediation hypotheses. Moreover, the option here is also justified as this work extends the mediation process to multiple independent variables.

## 4. Findings and Discussion

### 4.1. Sample Profile

The selection of this population was justified as it ensured a diversified sample with the representation of distinct municipal workers with distinct socio-economic characteristics, professional roles, education, contracts, and working in different sized municipal institutions. Due to limitations, in terms of data access, the convenience sample procedure tried to incorporate the maximum number of institutions, for a total number of 308 municipalities in Portugal, in order to ensure the total geographical coverage of Portugal, including five regions of continental Portugal: north, center, metropolitan area of Lisbon, Alentejo, and Algarve, and also two autonomous regions: Madeira and Azores. Based on the complete survey responses from 459 respondents, the convenience sample was generated. Previously, we described the socio-economic characteristics, professional profiles, and the size of the municipal employees’ institutions (see Table 1).

A total of 89.98% of participants reported a feeling of contributing to productivity, with 69.28% stating that they felt they were experiencing a good QWL. Nevertheless, only 27.67% of workers considered their salary appropriate. Taking as reference the three dimensions of assessing burnout, i.e., emotional exhaustion, cynicism, and low effectiveness, 44.23% agreed they experienced emotional exhaustion and feelings of high levels of stress/anxiety in the workplace; 28.54% expressed feelings of cynicism, stating they had become more critical at work, both of colleagues and working conditions; and 30.06% felt low effectiveness at work, lacking satisfaction in job achievements. Table 2, presented below, summarizes this descriptive information.

### 4.2. Burnout and QWL: Contribution to Productivity as Mediator (Model 1)

Concerning model 1, the direct effects (Table 3), indirect effects (Table 4), and total effects (Table 5) were assessed using one mediator variable, i.e., contribution to productivity. The level of z (and/or p) considered to be significant for the statistical analyses was 0.05. Regarding the first dimension of burnout, i.e., emotional exhaustion, the total effect is −0.8160712, which is the effect found if there is no mediator in the model. It is significant with a z of −4.79. The direct effect of emotional exhaustion is −0.8076825, which, while still significant (z = −4.74), is smaller than the total effect. The indirect effect of emotional exhaustion that passes through the contribution to productivity is −0.008388 and is not significant.

The proportion of the total mediated effect is −0.008388/−0.8160712, resulting in −0.8263497. The ratio of the indirect to direct effect is −0.008388/−0.8076825, equaling −0.0103852. The ratio of the total to direct effect corresponds to −0.8160712/−0.8076825, equaling −1.0103861. The proportion of the total effect that is mediated is almost −0.83, which is a respectable amount. The ratio of the indirect effect to the direct effect is about −0.010, and the total effect is about −1.010 times the direct effect.

Concerning the second dimension of burnout, i.e., cynicism, its total effect is −0.8564164, which is the effect found if there is no mediator in the model. It is significant, with a z of −5.00. The direct effect of cynicism is −0.1562514, which, while still significant (z = −1.95), is smaller than the total effect. The indirect effect of cynicism that passes through the contribution to productivity is −0.0116253 and is also not significant.

The proportion of the total effect mediated is −0.0116253/−0.8564164, resulting in −0.0135743. The ratio of the indirect to direct effect is −0.0116253/−0.8447911, equaling −0.0137611. The ratio of the total to direct effect corresponds to −0.8564164/−0.8447911, equaling −1.0137611. The proportion of the total effect that is mediated is almost −0.014, which is a small amount. The ratio of the indirect effect to the direct effect is about −0.014, and the total effect is about −1.014 times the direct effect.

Concerning the third dimension assessed for burnout, i.e., the feeling of low effectiveness, its total effect is −1.75465, which is the effect found if there is no mediator in the model. It is significant, with a z of −9.81. The direct effect of low effectiveness is −0.2946748, which, while still significant (z = −3.55), is smaller than the total effect. The indirect effect of low effectiveness that passes through the contribution to productivity is −0.0219241, and it is also not significant.

The proportion of the total effect mediated is −0.0219241/−1.75465, resulting in −0.0124948. The ratio of the indirect to direct effect is −0.0219241/−1.732726, equaling −0.0126529. The ratio of the total to direct effect corresponds to −1.75465/−1.732726, equaling −1.0126528. The proportion of the total effect that is mediated is almost -0.013, which is a small amount. The ratio of the indirect effect to the direct effect is about -0.013, and the total effect is about −1.013 times the direct effect.

### 4.3. Burnout and QWL: Contribution to Productivity and Appropriate Salary as Mediators (Model 2)

Considering model 2, the same set of effects are analyzed (cf. Table 6, Table 7 and Table 8), evaluating two mediators (i.e., contribution to productivity and appropriate salary). For the first dimension of burnout, i.e., emotional exhaustion, the total effect is −1.054325, which is the effect found if there are no mediators in the model. It is significant, with a z of −6.64. The direct effect for emotional exhaustion is -1.019637, which is still quite significant (z = −5.99), although slightly smaller than the total effect. The indirect effect of emotional exhaustion that passes through the contribution to productivity and appropriate salary is −0.0346888 and is not significant.

The proportion of the total effect mediated is −0.0346888/−1.054325, resulting in −0.0329014. The ratio of the indirect to direct effect is −0.0346888/−1.019637, equaling −0.0340207. The ratio of the total to direct effect corresponds to −1.054325/−1.019637, equaling −1.0340199. The proportion of the total effect that is mediated is almost −0.33, which is a respectable amount. The ratio of the indirect effect to direct effect is about −0.03, and the total effect is about −1.034 times the direct effect.

Concerning the second dimension of burnout, i.e., cynicism, its total effect is -1.011313, which is the effect found if there are no mediators in the model. It is significant, with a z of −6.30. The direct effect for cynicism is −0.9078191, which, while still significant (z = −5.29), is smaller than the total effect. The indirect effect of cynicism that passes through the contribution to productivity is −0.01034937 and is also not significant.

The proportion of the total effect mediated is −0.01034937/−1.011313, resulting in −0.0102335. The ratio of the indirect to direct effect is −0.01034937/−0.9078191, equaling −0.0114001. The ratio of the total to direct effect corresponds to −1.011313/−0.9078191, equaling −1.1140027. The proportion of the total effect that is mediated is almost −0.010, which is a small amount. The ratio of the indirect effect to direct effect is about -0.011, and the total effect is about −1.114 times the direct effect.

Regarding the third dimension of burnout, i.e., the feeling of low effectiveness, its total effect is −1.716576, which is the effect found if there are no mediators in the model. It is significant, with a z of −10.15. The direct effect for low effectiveness is −1.444998, which, while still significant (z = −8.13), is smaller than the total effect. The indirect effect of low effectiveness that passes through the contribution to productivity is −0.2715781, and it is also highly significant.

The proportion of the total effect mediated is −0.2715781/−1.716576, resulting in −0.1582091. The ratio of the indirect to direct effect is −0.2715781/−1.444998, equaling −0.1879435. The ratio of the total to direct effect corresponds to −1.716576/−1.444998, equaling −1.1879435. The proportion of the total effect that is mediated is almost −0.16, which is quite an important amount. The ratio of the indirect effect to direct effect is about −0.19, and the total effect is about −1.19 times the direct effect.

### 4.4. Discussion

The COVID-19 pandemic showed an increase of stress levels and emotional-exhaustion experiences related to employees’ occupational health, with higher absenteeism rates, loss of productivity, and increased healthcare consumption [7]. In fact, such impacts showed an important correlation with the loss of QWL and work ability [9].

The findings from this study show that, concerning the three dimensions of burnout assessment, i.e., emotional exhaustion, cynicism, and low effectiveness, approximately 44% answered that they were experiencing a condition of emotional exhaustion, feeling high stress/anxiety levels in the workplace. Almost 29% expressed feelings of cynicism, stating they had become more critical at work, both of colleagues and working conditions. For 30% of workers, feelings of being less effective at work and lacking satisfaction in job achievements occurred often.

Aligned with the results of prior research, COVID-19 not only impacted negatively on stress levels and QWL among frontline workers in the healthcare sector (physicians, nurses, and hospital workers) but also on other public servants [12,13]. Public servants closer to citizens, such as municipal workers, were suddenly forced to work remotely, changing public-sector work environments, raising contingent supervisory demands [12], and acting as a major stressor, increasing chronic anxiety, social isolation, and relationship difficulties [48].

The findings presented here confirm the burnout syndrome as a multidimensional phenomenon whose dimensions affect QWL in different ways. To understand how much of the measured effect of the independent variable (burnout dimensions) on the dependent variable (QWL) is attributable to motivating and economic mediator variables: contribution to productivity (in model 1) and satisfaction with salary (in model 2), we used mediation analysis. In model 1 (cf. Table 2, Table 3 and Table 4), the total effect comprises a direct effect pathway of the independent variables on the dependent variable (path C) and an indirect pathway of the independent variables on the dependent variable through the mediator (path A and path B). The first model analyzed the mediator effect of the contribution to productivity on the relation between the first dimension analyzed for burnout, i.e., emotional exhaustion, on QWL. Our results revealed a significant direct effect but found no significance in the mediating role of the worker feeling productive between this dimension of burnout and QWL. This mediating role had no significant effect concerning the second dimension, cynicism, or the third dimension, the feeling of low effectiveness, only showing important direct effects.

In contrast, model 2, assessing the mediation role of workers’ contribution to productivity if associated with lower satisfaction with salary in the relation between municipal workers’ burnout and decreased QWL (cf. Table 5, Table 6 and Table 7), our findings confirm the significant direct effects, for the first two dimensions of burnout on QWL but, as in model 1, found no significance in the mediating role of the worker feeling productive and satisfied with salary between this dimension of burnout and QWL. Nevertheless, and contrasting with the first dimensions of the burnout syndrome, the third one, i.e., the sense of low effectiveness, revealed not only important direct effects but highly significant indirect effects, which are in line with prior research highlighting the multidimensionality of burnout and the diverse effects on QWL.

The results of the present study are aligned with the literature on the determinants of QWL, which highlights the association between altered work conditions, including for public servants not directly working in health-related areas, and increased worker burnout exacerbated by the consequences for families’ personal life and budget [47].

## 5. Conclusions

This study contributes to the literature on occupational health, with an application for the public sector, especially the situation of municipal workers developing professional activities during the COVID-19 pandemic. It addressed the relationship between the multidimensional construct of burnout and QWL, by innovating in empirical assessment, considering both the isolated mediator of contribution to productivity and the joint mediator of the contribution to productivity and an appropriate salary.

In fact, the set of evidence presented outlines an important mediating role of not feeling contributive at work, as well as not being well paid, in the link between the burnout syndrome dimension of low effectiveness and QWL. These findings shed new light on the need for public managers in leading roles, as well as human resource departments in the public sector, to design innovative incentives, appropriate measures, and work schedules to counterbalance such tendencies.

Regarding implications for public authorities, and based on our previous direct contacts and auscultation in the context of online training delivered to municipal workers, in order to address the negative association between the three burnout dimensions: emotional exhaustion, feelings of cynicism, and feeling less effective, and the motivating and economic mediators: contribution to productivity and appropriate salary; progressive remuneration mechanisms should be created, including a (fixed) basic salary and a (variable) supplementary salary indexed to productivity goals, and aligned with a new production and evaluation framework, to be implemented in public administration. Also taking into consideration the results of brainstorming, co-creation, and open innovation groups dynamics promoted in the previously referred online training delivered to municipal workers, further action is required to promote occupational health, QWL, and gender equality in public administration, aiming to diminish income inequality and foster the work–life balance of both female and male public servants.

Adding to the above, and considering the previous experience and project-based and organization innovation views of the authors collected in the previous exercise of public administration roles, it is suggested that public managers adopt innovative organizational practices, including flexible work schedules; project management practices; work innovation labs; mindfulness programs; hybrid work formats; and special incentives for workers with families, including children at school and older people with special needs, in order to tackle the negative associations and mediating effects found in the current study.

Despite the important set of discoveries made in the course of the current study, it is not free of limitations that should be considered in future research. Firstly, this study uses a cross-sectional online survey, which might have limited access by municipal workers who are less familiar with the internet or less likely to use it, for instance, manual workers rather than office workers. The convenience sampling procedure was obtained by direct contact with chief positions (middle managers) in the municipalities during online training courses, something that could bias access by some groups or individuals. This is a non-probability sampling method wherein units are selected for inclusion in the sample because they are the easiest for the researcher to access. The use of this non-random sampling can be due to availability at a given time or willingness to participate in the research, as occurred in the context of the previously referred online training.

One limitation associated with the empirical test of the convenience sample is that mixing functions and qualifications of municipal workers could bias the results, making it impossible to focus on one category of municipal workers. Nevertheless, bearing in mind the sample’s dimension and the observed diversified set of institutional roles, it was found to be advantageous to consider municipal workers with leadership roles, as well as qualified and non-qualified workers.

It is also worthwhile to outline that the limited sample size was confined to Portuguese municipalities and, therefore, these results cannot be generalized to municipalities in other countries. Another aspect associated with the representativeness is the fact that the sample is related only to public municipalities, excluding from the study other public municipal firms. Therefore, representativeness is limited, and the results of the study cannot be generalized to the entire Portuguese municipal system, including the entrepreneurial units with the participation of municipal bodies.

The study was carried out during a five-month period, during a pandemic, which corresponded to one lockdown and then the relaxation of some lockdown measures. Furthermore, the inexistence of data gathered before the pandemic means comparisons cannot be made. Future studies could focus on participants who already felt burnout symptoms before the COVID-19 pandemic, to examine the increase, maintenance, or decrease of symptoms during the pandemic. When gathering information through a questionnaire, bias can appear linked to the tendency to present a favorable image of oneself. This tendency could be increased as sending the invitations to the leading positions inside the institutions can also produce a biased collection. In the sample used, 26% have leadership positions and 65% were qualified workers, while only 8% were non-qualified workers.

Future research endeavors on the relationship between burnout and QWL should perform a comparative analysis of different hierarchical positions, for different activities in the public sector, in order to assess the mediating effects of the contribution to productivity and appropriate salary, considering different cohorts of remuneration, education, and age.

## Figures and Tables

**Figure 1 ijerph-19-13035-f001:**
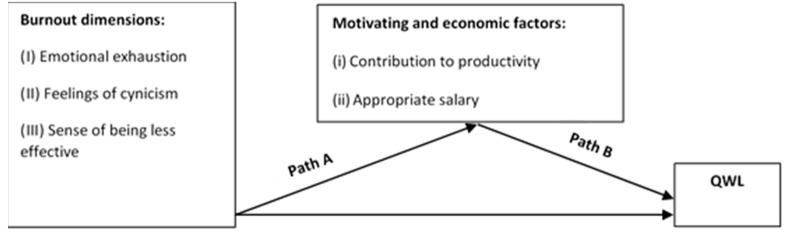
Mediators of burnout and QWL: Model design.

**Table 1 ijerph-19-13035-t001:** Socio-economic characteristics, professional profiles, and the size of the institutions of the municipal workers.

Socio-Economic Characteristics		
Gender	Female	67.00%
	Male	33.00%
Age	20–25 years	0.60%
	26–35 years	8.00%
	36–45 years	34.00%
	46–55 years	38.00%
	>55 years	19.40%
Age	Single	19.00%
	Married	67.00%
	Other	14.00%
**Professional profiles**		
Insitutional role	Leadership positions	25.60%
	Qualified workers	66.40%
	Non-qualified workers	8.00%
Education	Post-graduate	24.00%
	College	48.00%
	Secondary school	28.00%
Contract	Permanent	50.00%
	Open	40.00%
	Fixed period	5.00%
	Temporary	5.00%
**Municipal Institutions**		
Size	<50 employees	5.00%
	[50; 249] employees	10.00%
	[250; 999] employees	36.00%
	≥1000 employees	49.00%

**Table 2 ijerph-19-13035-t002:** Sample characteristics (n = 459).

Characteristics	n	%
Contribution to productivity	413	89.98
QWL	318	69.28
Appropriate salary	127	27.67
Emotional exhaustion	203	44.23
Cynicism	131	28.54
Low effectiveness	138	30.06

**Table 3 ijerph-19-13035-t003:** Model 1: Direct effects.

OIM	Coef.	Std.Err.	z	P > z	[95%	Conf.Int]
Structural						
contribution to productivity						
QWL	0.0744012	0.099308	0.75	0.454	−0.1202389	0.2690412
Burnout Dimension I						
Emotional-exhaustion	<-					
contribution to productivity	−0.1127498	0.0799756	−1.41	0.159	−0.2694992	0.0439995
QWL	−0.8076825 ***	0.1702603	−4.74	0.000	−1.141387	−0.4739784
Burnout Dimension II						
Cynicism	<-					
contribution to productivity	−0.1562514 *	0.0802144	−1.95	0.051	−0.3134687	0.000966
QWL	−0.8447911 ***	0.1707687	−4.95	0.000	−1.179492	−0.5100907
Burnout Dimension III						
Low-effectiveness	<-					
contribution to productivity	−0.2946748 ***	0.0829586	−3.55	0.000	−0.4572706	−0.1320789
QWL	−1.732726 ***	0.1766108	−9.81	0.000	−2.078877	−1.386575

Legend: *** 1% statistical significance; and * 10% statistical significance.

**Table 4 ijerph-19-13035-t004:** Model 1: Indirect effects.

OIM	Coef.	Std.Err.	z	P > z	[95%	Conf.Int]
Structural						
contribution to productivity						
QWL	0	(no	path)			
Burnout Dimension I						
Emotional-exhaustion	<-					
contribution to productivity	0	(no	path)			
QWL	−0.0083887	0.0126798	−0.66	0.508	−0.0332407	0.0164633
Burnout Dimension II						
Cynicism	<-					
contribution to productivity	0	(no	path)			
QWL	−0.0116253	0.0166251	−0.70	0.484	−0.0442099	0.0209594
Burnout Dimension III						
Low-effectiveness	<-					
contribution to productivity	0	(no	path)			
QWL	−0.0219241	0.0299074	−0.73	0.464	−0.0805416	0.0366933

**Table 5 ijerph-19-13035-t005:** Model 1: Total effects.

OIM	Coef.	Std.Err.	z	P > z	[95%	Conf.Int]
Structural						
contribution to productivity						
QWL	0.0744012	0.099308	0.75	0.454	−0.1202389	0.2690412
Burnout Dimension I						
Emotional-exhaustion	<-					
contribution to productivity	−0.1127498	0.0799756	−1.41	0.159	−0.2694992	0.0439995
QWL	−0.8160712 ***	0.1705243	−4.79	0.000	−1.150293	−0.4818497
Burnout Dimension II						
Cynicism	<-					
contribution to productivity	−0.1562514 *	0.0802144	−1.95	0.051	−0.3134687	0.000966
QWL	−0.8564164 ***	0.1713683	−5.00	0.000	−1.192292	−0.5205407
Burnout Dimension III						
Low-effectiveness	<-					
contribution to productivity	−0.2946748 ***	0.0829586	−3.55	0.000	−0.4572706	−0.1320789
QWL	−1.75465 ***	0.1789124	−9.81	0.000	−2.105312	−1.403988

Legend: *** 1% statistical significance; and * 10% statistical significance.

**Table 6 ijerph-19-13035-t006:** Model 2: Direct effects.

OIM	Coef.	Std.Err.	z	P > z	[95%	Conf.Int]
Structural						
contribution to productivity						
QWL	0.2031458 **	0.0940108	2.16	0.031	0.018888	0.3874035
Appropriate salary	<-					
QWL	1.210284 ***	0.1492337	8.11	0.000	0.9177917	1.502777
Burnout dimension I						
Emotional-exhaustion	<-					
contribution to productivity	−0.0748569	0.0787106	−0.95	0.342	−0.2291269	0.0794131
Appropriate salary	−0.016097	0.0495843	−0.32	0.745	−0.1132804	0.0810865
QWL	−1.019637 ***	0.1703282	−5.99	0.000	−1.353474	−0.6857996
Burnout dimension II						
Cynicism	<-					
contribution to productivity	−0.1226739	0.0793634	−1.55	0.122	−0.2782234	0.0328755
Appropriate salary	−0.0649211	0.0499955	−1.30	0.194	−0.1629105	0.0330683
QWL	−0.9078191 ***	0.1717408	−5.29	0.000	−1.244425	−0.5712133
Burnout Dimension III						
Low-effectiveness	<-					
contribution to productivity	−0.2445894 ***	0.0820851	−2.98	0.003	−0.4054733	−0.0837055
Appropriate salary	−0.1833378 ***	0.0517101	−3.55	0.000	−0.2846877	−0.0819878
QWL	−1.444998 ***	0.1776306	−8.13	0.000	−1.793147	−1.096848

Legend: *** 1% statistical significance; and ** 5% statistical significance.

**Table 7 ijerph-19-13035-t007:** Model 2: Indirect effects.

OIM	Coef.	Std.Err.	z	P > z	[95%	Conf.Int]
Structural						
contribution to productivity						
QWL	0	(no	path)			
Appropriate salary	<-					
QWL	0	(no	path)			
Burnout dimension I						
Emotional-exhaustion	<-					
contribution to productivity	0	(no	path)			
Appropriate salary	0	(no	path)			
QWL	−0.0346888	0.0627665	−0.55	0.580	−0.1577088	0.0883312
Burnout dimension II						
Cynicism	<-					
contribution to productivity	0	(no	path)			
Appropriate salary	0	(no	path)			
QWL	−0.1034937	0.0646212	−1.60	0.109	−0.2301489	0.0231615
Burnout Dimension III						
Low-effectiveness	<-					
contribution to productivity	0	(no	path)			
Appropriate salary	0	(no	path)			
QWL	−0.2715781 ***	0.0741744	−3.66	0.000	−0.4169573	−0.126199

Legend: *** 1% statistical significance.

**Table 8 ijerph-19-13035-t008:** Model 2: Total effects.

OIM	Coef.	Std.Err.	z	P > z	[95%	Conf.Int]
Structural						
contribution to productivity						
QWL	0.2031458 **	0.0940108	2.16	0.031	0.018888	0.3874035
Appropriate salary	<-					
QWL	1.210284 ***	0.1492337	8.11	0.000	0.9177917	1.502777
Burnout dimension I						
Emotional-exhaustion	<-					
contribution to productivity	−0.0748569	0.0787106	−0.95	0.342	−0.2291269	0.0794131
Appropriate salary	−0.016097	0.0495843	−0.32	0.745	−0.1132804	0.0810865
QWL	−1.054325 ***	0.1586904	−6.64	0.000	−1.365353	−0.743298
Burnout dimension II						
Cynicism	<-					
contribution to productivity	−0.1226739	0.0793634	−1.55	0.122	−0.2782234	0.0328755
Appropriate salary	−0.0649211	0.0499955	−1.30	0.194	−0.1629105	0.0330683
QWL	−1.011313 ***	0.1605389	−6.30	0.000	−1.325963	−0.6966623
Burnout Dimension III						
Low-effectiveness	<-					
contribution to productivity	−0.2445894 ***	0.0820851	−2.98	0.003	−0.4054733	−0.0837055
Appropriate salary	−0.1833378 ***	0.0517101	−3.55	0.000	−0.2846877	−0.0819878
QWL	−1.716576 ***	0.1691313	−10.15	0.000	−2.048067	−1.385085

Legend: *** 1% statistical significance; and ** 5% statistical significance.

## Data Availability

The data presented in this study are available on request from the corresponding author.

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
