# Peer review of "Burnout and Quality of Work Life among Municipal Workers: Do Motivating and Economic Factors Play a Mediating Role?"

_ijerph, 2022, doi:10.3390/ijerph192013035_

Round 1
Reviewer 1 Report
I would like to thank the authors for this research that aims analyze the relationship between burnout and quality of work life among 15 public servants subjected to higher levels of stress and emotional exhaustion, interacting with their 16 occupational health in the context of the COVID 19 pandemic.
The study aims to answer this question: Does the feeling of contributing to productivity and receiving an appropriate salary mediate the relationship between the burnout condition and the QWL of municipal workers in the context of the Covid-19 pandemic?
The research subject is timely, innovative, and highly interesting. It also fits the aim and scope of the journal.
The research is well designed and follows a sound scientific research method.
Results are clear and could have an impact among the community of researchers.
Conclusion, implications and limitations are clearly described.
However, the research needs minor adjustments:
The title needs adjustment:
The research was focusing only on municipal workers. It is more appropriate to reflect that in the title.
Abstract:
The research focused on municipal workers in Portugal. To be more precise, it would be better to change these public servants by municipal workers.
Line 159: Remote work was a challenge that has negative effects but also many positive effects. Research showed also the positive outcomes of remote work.
Research methods: Paragraph 3.2: To simplify reading, it would be better to support by a table.
Lines 267-268: Mixing functions and qualifications of municipal workers could bias the results. It was better to focus on one category of municipal workers. You need to highlight that in the limitations.
You can diversify your sources and solidify your finding by recent references.
Other minor comments are directly attached to the manuscript.

Author Response
Reviewer 1:
Comment 1.1:
I would like to thank the authors for this research that aims analyze the relationship between burnout and quality of work life among 15 public servants subjected to higher levels of stress and emotional exhaustion, interacting with their 16 occupational health in the context of the COVID 19 pandemic.
The study aims to answer this question: Does the feeling of contributing to productivity and receiving an appropriate salary mediate the relationship between the burnout condition and the QWL of municipal workers in the context of the Covid-19 pandemic?
The research subject is timely, innovative, and highly interesting. It also fits the aim and scope of the journal.
The research is well designed and follows a sound scientific research method.
Results are clear and could have an impact among the community of researchers.
Conclusion, implications and limitations are clearly described.
Authors’ Response 1.1: Thanks to the reviewer for such kind and positive appreciation.
Comment 1.2:
However, the research needs minor adjustments:
The title needs adjustment:
The research was focusing only on municipal workers. It is more appropriate to reflect that in the title.
Authors’ Response 1.2: Thanks to the reviewer for such appreciation. As per valued observation of the reviewer, the authors have revised the title of the manuscript accordingly, as well as several expressions used along the manuscript, outlining the focus of the current study on municipal workers.
Comment 1.3:
Abstract:
The research focused on municipal workers in Portugal. To be more precise, it would be better to change these public servants by municipal workers.
Authors’ Response 1.3: Thanks to the reviewer for such appreciation. As per valued observation of the reviewer, the authors have revised the abstract accordingly.
Comment 1.4:
Line 159: Remote work was a challenge that has negative effects but also many positive effects. Research showed also the positive outcomes of remote work.
Authors’ Response 1.4: Thanks to the reviewer for such appreciation. As per valued observation of the reviewer, the authors revised the content of section 2. Burnout during COVID-19 for public servants, inserting in line 158, the new following sentence:
Nevertheless, the same authors outline that the isolation of workers at home had mixed effects on adult health behaviors in China, stressing that those workers focused more on their eating quality and patterns, which had a positive influence on their quality of life.
Comment 1.5:
Research methods: Paragraph 3.2: To simplify reading, it would be better to support by a table.
Authors’ Response 1.5: Thanks to the reviewer for such appreciation. As per valued observation of the reviewer, the authors introduced a new summing-up Table 1, concerning the socio-economic characteristics and professional profiles of municipal workers developing professional activities during the COVID-19 pandemic, in municipal institutions with different sizes.
Comment 1.6:
Lines 267-268: Mixing functions and qualifications of municipal workers could bias the results. It was better to focus on one category of municipal workers. You need to highlight that in the limitations.
Authors’ Response 1.6: Thanks to the reviewer for such appreciation. As per valued observation of the reviewer, the authors introduced the following sentence in the last section: 5. Concluding remarks and implications, addressing the limitation pointed out:
One limitation associated with the empirical test of the convenience sampling procedure is that mixing functions and qualifications of municipal workers could bias the results, making also impossible to focus on one category of municipal workers. Nevertheless, bearing in mind the sample’s dimension and the observed diversified set of institutional roles, it was found to be advantageous to consider municipal workers with leadership roles, as well as qualified and non-qualified workers.
Comment 1.7:
You can diversify your sources and solidify your finding by recent references.
Authors’ Response 1.7: Thanks to the reviewer for such appreciation. As per valued observation of the reviewer, the authors introduced several recent references, namely:
In fact, women working from home during COVID-19 lockdown, experienced a set of negative effects from working remotely, namely the invasion of private, family time, the incurrence in distractions while working deriving in failures, and the increasing interference of work in private life (3).
Conversely, not only at the employees’ level, but also at the level of SME owners, who faced during COVID-19 crisis, high stress, mostly derived from the shortage of personnel, financial constraints, liquidities, repeated closures, and reopening, as well as a high difficulty in adapting to such changing environment (8).
Focusing on younger, women, unmarried healthcare workers, with low monthly in-comes, studied in Jordan during the pandemic, revealed high anxiety symptoms, being exacerbated one year after the beginning of COVID-19, particularly for the physicians, with intense schedules and that got infected (17). This situation is intensified in the public health care system, during pandemics, as it is characterized by a scarcity of resources, reduced accountability, thus increasing the lack of trustworthiness in the health system (18). Strategies to improve health care systems’ efficiency, physicians’ motivation, management routines, patient flows and information are a need during extreme crisis, such as COVID-19. These strategies were implemented in a nodal-designated COVID-19 center in Qatar and proved to be effective, lowering mortality, increasing efficacy, capabilities, and patient satisfaction (19).
In addition, fear of exposure to COVID-19 or even due to the scarcity of personal protective equipment (PPE), allocation of resources, accountability, and efficient management of patients’ flows, as well as lack or reduced training on practical skills to deal with the emergency situations and critical care, can put even more pressure on front line/health professionals (18,44).
Comment 1.8:
Other minor comments are directly attached to the manuscript.
Authors’ Response 1.8: Thanks to the reviewer for such appreciation and highly valuable feedback, which was taken into consideration in the revised version of the manuscript.

Reviewer 2 Report
Thank you for submitting your paper on assessing burnout and quality of work life (QWL) in Portuguese public servants during defined periods of the COVID-19 pandemic. The paper reads well in the main, but I have made some edits and corrections in terms of grammar and syntax, for your review. Additionally I have asked some questions in terms of the scientific content and methodology. All my changes and comments have been added as sticky notes in the original PDF version of the manuscript, attached herewith.
1) Is the participant number (N=459) a representative sample of the public service population in Portugal? I anticipate the sample size is very small. Please discuss the impacts of a 'too small' sample size on the statistical analyses (and outcomes) of your study (this could be a limitation of the study).
2) What was the response rate for the questionnaire? I.e.. 459 responded out of a total population of how many?
3) The socio-demographic characteristics of the sample should be presented in a table format (not free text) to make the data easier and more interesting to read.
4) Section 2 of the paper (still part of the background to the study) is very long, and somewhat repetitive. The greater emphasis should be on methods, data analyses, findings and discussion. Section 2 should be summarized and streamlined. You can still use all the references as they are appropriate, but can for example, omit the very specific details of the various studies that are cited.
5) Please correct some minor typos in the reference list, as well as an access error in one of the URL links.

Author Response
Reviewer 2:
Comment 2.1:
Thank you for submitting your paper on assessing burnout and quality of work life (QWL) in Portuguese public servants during defined periods of the COVID-19 pandemic. The paper reads well in the main, but I have made some edits and corrections in terms of grammar and syntax, for your review. Additionally, I have asked some questions in terms of the scientific content and methodology. All my changes and comments have been added as sticky notes in the original PDF version of the manuscript, attached herewith.
Authors’ Response 2.1: Thanks to the reviewer for such appreciation and highly valuable feedback, which was taken into consideration in the revised version of the manuscript.
Comment 2.2:
1) Is the participant number (N=459) a representative sample of the public service population in Portugal? I anticipate the sample size is very small. Please discuss the impacts of a 'too small' sample size on the statistical analyses (and outcomes) of your study (this could be a limitation of the study).
Authors’ Response 2.2: Thanks to the reviewer for such appreciation. As per valued observation of the reviewer, the authors introduced new sentences justifying the use of a convenience sample procedure, as well as presenting the limitations, as follows:
In section 3.3. Survey procedures, the following sentence was added:
With the motivation of accomplishing the objectives, this study was analytical and correlational, because it sought to explore the variables and the relationships between them, and it was cross-sectional because the sample was collected in a single period. The purpose of the study was descriptive because it aimed to analyze the relationship between burnout and quality of work life among municipal workers subjected to higher levels of stress and emotional exhaustion, interacting with their occupational health in the context of the COVID 19 pandemic. Through a quantitative, objectivist and, therefore, deductive approach, this research was supported by models built on results and previous research, with quantitative indicators collected through a questionnaire.
In section 4.1. Survey procedures, the following sentences were added:
The selection of this population was justified as it ensured a diversified sample with the representation of distinct municipal workers with distinct socio-economic characteristics, professional roles, education, contract and working in different sized municipal institutions. Due to limitations, in terms of data access, the convenience sample procedure tried to incorporate the maximum number of institutions, for the total number of 308 municipalities in Portugal, in order to ensure the total geographical coverture of Portugal, including five regions from continental Portugal: North, Centre, Metropolitan area of Lisbon, Alentejo, and Algarve, and also two autonomous regions: Madeira and Azores.
Based on the complete survey responses from 459 respondents, the convenience sample was generated. Previously, we described the socio-economic characteristics, professional profiles, and size of municipal employees' institutions (see Table 1).
In section 5. Concluding remarks and implications, the following sentence was added:
It is also worthwhile to outline that the limited sample size was confined to Portuguese Municipalities and, therefore, these results cannot be generalized to Municipalities in other countries. Another aspect associated to the representativeness is the fact that the sample is related only to public municipalities, excluding from the study other public municipal firms. Therefore, representativeness is limited, and the results of the study cannot be generalized to the entire Portuguese Municipal system, including the entrepreneurial units with the participation of Municipal bodies.
Comment 2.3:
2) What was the response rate for the questionnaire? I.e.. 459 responded out of a total population of how many?
Authors’ Response 2.3:
Thanks to the reviewer for such appreciation. As per valued observation of the reviewer, and considering the previous question 2.2., due to limited acess to municipal workers involved in the development of professional activities during the COVID-19 pandemic, the authors aim to justify the use of convenience sample and described in section 4.1. Survey procedures, the procedure followed, in the following terms:
Based on the complete survey responses from 459 respondents, the convenience sample was generated. Previously, we described the socio-economic characteristics, professional profiles, and size of municipal employees' institutions (see Table 1).
In addition, as stated in the previous answer, the limited sample size is a limitation that, in fact, prevents the generalization of results.
Comment 2.4:
3) The socio-demographic characteristics of the sample should be presented in a table format (not free text) to make the data easier and more interesting to read.
Authors’ Response 2.4:
Thanks to the reviewer for such appreciation. As per valued observation of the reviewer, the authors introduced a new summing-up Table 1, concerning the socio-economic characteristics and professional profiles of municipal workers developing professional activities during the COVID-19 pandemic, in municipal institutions with different sizes.
Comment 2.5:
4) Section 2 of the paper (still part of the background to the study) is very long, and somewhat repetitive. The greater emphasis should be on methods, data analyses, findings, and discussion. Section 2 should be summarized and streamlined. You can still use all the references as they are appropriate, but can for example, omit the very specific details of the various studies that are cited.
Authors’ Response 2.5:
Thanks to the reviewer for such appreciation. As per valued observation of the reviewer, the authors downsized and streamlined Section 2, as recommended. Although it should be noted that following the recommendations of Reviewer 1, 5 new references were added to this section.
Comment 2.6:
5) Please correct some minor typos in the reference list, as well as an access error in one of the URL links.
Authors’ Response 2.6:
Thanks to the reviewer for such appreciation. As per valued observation of the reviewer, the authors revised all the typos and errors, as recommended.

Reviewer 3 Report
A well-written and scientifically-sound manuscript. The authors addressed the factors underlying burnout in the workplace as a by-product of the COVID-19 pandemic. The literature review was profound and critical of how "the feeling of contributing" played a role in workplace burnout. However, I am curious to read more on the "dissatisfactory/satisfactory" salary ranges to see if there is a consistent range across different positions or change accordingly (especially when salary is a significant factor in burnout, as shown in the statistical results). I also appreciate the authors' efforts in explaining the instrument (i.e., a survey built upon established ones) and the results in either writing or tables. The authors might need to elaborate more on Section 5. Conclusions and Implications. For example, is there any literature review that supports the implications for public authorities? Alternatively, at least an explanation of how the authors come up with the suggestions. My final comment is that while multiple researchers address the topic, this manuscript is of another caliber.
Author Response
Reviewer 3:
Comment 3.1:
A well-written and scientifically-sound manuscript. The authors addressed the factors underlying burnout in the workplace as a by-product of the COVID-19 pandemic. The literature review was profound and critical of how "the feeling of contributing" played a role in workplace burnout. However, I am curious to read more on the "dissatisfactory/satisfactory" salary ranges to see if there is a consistent range across different positions or change accordingly (especially when salary is a significant factor in burnout, as shown in the statistical results). I also appreciate the authors' efforts in explaining the instrument (i.e., a survey built upon established ones) and the results in either writing or tables.
Authors’ Response 3.1: Thanks to the reviewer for such kind and positive appreciation. We will keep in mind your highly valuable comment on the role played different positions on "dissatisfactory/satisfactory" salary ranges, in a future research, including a large sample size or cross-country samples, for testing professional groups with distinct institutional roles, which unfortunately in the scope of the current study is not possible to test.
Comment 3.2:
The authors might need to elaborate more on Section 5. Conclusions and Implications. For example, is there any literature review that supports the implications for public authorities? Alternatively, at least an explanation of how the authors come up with the suggestions.
Authors’ Response 3.2: Thanks to the reviewer for such kind and positive appreciation. As per valued observation of the reviewer, since at the level of our current knowledge, there is an absence of previous literature focusing on the promotion of quality of work life in the context of municipal workers developing professional activities during the COVID-19 pandemic, the authors tried to to justify the suggestions, as recommended. The following sentences were introduced in the final section 5. Concluding remarks and implications:
Regarding implications for public authorities, and based in our previous direct contacts and auscultation in the context of online training delivered to municipal workers, in order to address the negative association between the three burnout dimensions: emotional exhaustion; feelings of cynicism; feeling less effective; and the motivating and economic mediators: contribution to productivity; and appropriate salary; progressive remuneration mechanisms should be created, including a (fixed) basic salary and a (variable) supplementary salary indexed to productivity goals, and aligned with a new production and evaluation framework, to be implemented in public administration. Also taking into consideration the results of brainstorming, co-creation, and open innovation groups dynamics promoted in the previously referred online training delivered to municipal workers, further action is required to promote occupational health, QWL, and gender equality in public administration, aiming to diminish income inequality and foster the work life balance of both female and male public servants.
Adding to the above, and considering the previous experience and project-based and organization innovation views of the authors collected in the previous exercise of public administration roles, it is suggested that public managers adopt innovative organizational practices, including flexible work schedules, project management practices, work innovation labs, mindfulness programmes, hybrid work formats, special incentives for workers with families, including children at school and older people with special needs, in order to tackle the negative associations and mediating effects found in the current study.
Despite the important set of discoveries made in the course of the current study, it is not free of limitations that should be considered in future research. Firstly, this study uses a cross-sectional online survey, which might have limited access by municipal workers who are less familiar with the internet or less likely to use it, for instance, manual workers rather than office workers. The convenience sampling procedure was obtained by direct contact with chief positions (middle managers) in the municipalities during online training courses, something that could bias access by some groups or individuals.
Comment 3.3:
My final comment is that while multiple researchers address the topic, this manuscript is of another caliber.
Authors’ Response 3.3: Thanks to the reviewer for such kind and positive appreciation.
